# Separation and Analytical Techniques Used in Snake Venomics: A Review Article

Christina Sahyoun [1,2] , Mohamad Rima [1,†] , César Mattei [2] , Jean-Marc Sabatier [3] , Ziad Fajloun [1,4,*] and Christian Legros [2,*]

1 Laboratory of Applied Biotechnology (LBA3B), Azm Center for Research in Biotechnology and Its Applications, EDST, Lebanese University, Tripoli 1300, Lebanon; christina.sahyoun1@gmail.com (C.S.); mohamad.rima@hotmail.com (M.R.)

2 Univ Angers, INSERM, CNRS, MITOVASC, Team 2 CarMe, SFR ICAT, 49000 Angers, France; cesar.mattei@univ-angers.fr

3 Institut de Neurophysiopathologie (INP), Aix-Marseille Université CNRS, 13385 Marseille, France; sabatier.jm1@libertysurf.fr

4 Department of Biology, Faculty of Sciences III, Campus Michel Slayman Ras Maska, Lebanese University, Tripoli 1352, Lebanon

* Correspondence: ziad.fajloun@ul.edu.lb (Z.F.); christian.legros@univ-angers.fr (C.L.)

† Current address: StarkAge Therapeutics, Campus de l'Institut Pasteur de Lille, 59000 Lille, France.

**Abstract:** The deleterious consequences of snake envenomation are due to the extreme protein complexity of snake venoms. Therefore, the identification of their components is crucial for understanding the clinical manifestations of envenomation pathophysiology and for the development of effective antivenoms. In addition, snake venoms are considered as libraries of bioactive molecules that can be used to develop innovative drugs. Numerous separation and analytical techniques are combined to study snake venom composition including chromatographic techniques such as size exclusion and RP-HPLC and electrophoretic techniques. Herein, we present in detail these existing techniques and their applications in snake venom research. In the first part, we discuss the different possible technical combinations that could be used to isolate and purify SV proteins using what is known as bioassay-guided fractionation. In the second part, we describe four different proteomic strategies that could be applied for venomics studies to evaluate whole venom composition, including the mostly used technique: RP-HPLC. Eventually, we show that to date, there is no standard technique used for the separation of all snake venoms. Thus, different combinations might be developed, taking into consideration the main objective of the study, the available resources, and the properties of the target molecules to be isolated.

**Keywords:** separation techniques; analytical techniques; snake venom; biomolecules; bioassay-guided fractionation

## 1. Introduction

Venomous animals are widely spread all over the world. Accordingly, animal–human encounters are frequent in endemic regions, possibly ending up with human envenomation. Animal venoms induce different types of symptoms, leading sometimes to death [1]. Snake venoms (SVs) are the highlights of research owing to their fascinating complexity and their deadly effects [2]. Snakebite envenoming is a global health issue affecting up to 2.7 million people per year. The death toll is estimated to reach 138,000 deaths per year (https://www.who.int/news-room/fact-sheets/detail/snakebite-envenoming (accessed on 23 January 2022)). Due to their highly diverse composition, SVs cause a variety of clinical manifestations ranging from local tissue necrosis to systemic cardiovascular and neurologic symptoms. Nowadays, the single efficient treatment for snake envenomation is antivenom. Yet, the development of effective antivenoms is hampered by the complexity and the limited

understanding of the composition and biological activities of SVs, among others [3]. In addition to the appropriate development of antivenoms, a detailed characterization of SV components will help in the interpretation of clinical symptoms. It will also pave the way for the discovery of novel biomolecules with therapeutic interest [4]. To date, various drugs originating from SVs have been approved for clinical use. The most prominent drug Captopril$^{TM}$, an angiotensin-converting enzyme inhibitor, is indicated for the treatment of hypertension. It was developed from a bradykinin-potentiating enzyme isolated from *Bothrops jararaca* venom [5]. Aggrastat$^{TM}$ (Echistatin), another medication developed from *Echis carinatus* SV, is approved for the prevention of thrombotic complications. Reptilase isolated from different venoms of *Bothrops* species is approved for use during surgery for the prophylaxis and treatment of hemorrhage [6]. Accordingly, the exploration of SV components is of high significance to the medical field [7].

In fact, proteins and peptides constitute 90 to 95% of SV dry weight. They might contain more than 100 proteins mostly belonging to around 8–11 protein families [8]. Consequently, SVs are highly heterogeneous and require robust techniques to separate and identify protein isoforms. Recent advances in protein separation techniques provided means with higher resolution to evaluate these extremely heterogeneous mixtures. The separation of venom components is the first step in the process of venom decomplexation and analysis. It allows the repartition of molecules based on different biological, chemical, and physical properties [9]. Several techniques are currently available, but there is no standard technique used for the separation of SV proteins. Since they are relatively scarce and cannot be wasted to optimize separation techniques, an overview of previously used separation strategies is crucial. Accordingly, we present in this review the most used separation techniques currently available for the analysis of snake venoms. We then detail their implementation and use for different study purposes. In fact, it is the study objective that drives the choice of separation technique to be used. Therefore, we detail at first in this review the combinations of chromatographic techniques that are used to isolate and purify different snake venom proteins. On the other hand, we detail different workflows using different separation techniques for the evaluation of whole snake venom proteomic composition and distribution of SV protein families.

## 2. Methods Used for Separation of Venom Complex Mixtures

### 2.1. Chromatographic Techniques

Chromatographic methods are mostly employed for the separation of SVs and are preferred over electrophoretic techniques since they provide better resolution [10]. Liquid chromatography (LC) involves the partition of molecules between two phases: a stationary and a mobile phase. Molecules with different biochemical and physical features will interact differently with the stationary phase and thus will be separated from other components of the mixture [11]. Different types of chromatography are currently used for the separation of SVs, namely (i) size exclusion chromatography, (ii) ion-exchange chromatography, (iii) affinity chromatography and (iv) reversed-phase chromatography. A single chromatographic step is usually insufficient to isolate a molecule or reveal composition of the complex SV mixture. Hence, multiple chromatographic steps can be coupled together, which is known as multidimensional chromatography. Additionally, LC might be coupled with electrophoretic techniques for an improved protein isolation.

#### 2.1.1. Size Exclusion Chromatography

Size exclusion chromatography (SEC), also known as gel filtration chromatography, intends to separate mixtures of molecules based on their size (Figure 1(Aa)). SEC is a highly versatile technique where multiple factors could be modified to obtain the optimal separation of the target proteins. Numerous column matrices are available that are made of a single type of polymer or a combination of different types. The matrices used most frequently for the separation of SVs include dextran (Sephadex), dextran–polyacrylamide (Sephacryl) and dextran–agarose (Superdex). In addition to the choice of matrix, other

column parameters might be modified such as length and inner diameter that are directly related to the column fractionation range and resolution. Thus, if a high-resolution fractionation of molecule is required, then a narrow molecular weight (MW) range column should be used, covering the MW of the molecule. On the other hand, if whole venom fractionation is the objective, then selecting a column covering the molecular weight of all SV proteins is preferable [12]. Notably, SV proteins are distributed over a wide range of molecular weights (1.6–250 kDa); thus, columns with different molecular ranges (ranging from 1.5 to 1500 kDa) are used comprising Sephadex G50, Sephadex G75, Superdex G75, Superdex 200 and Sephacryl S-300. In addition to the choice of a column, the mobile phase selection is of high importance in every chromatographic process. Therefore, in SEC, a mobile phase with good ionic properties should be chosen to (i) minimize any type of interaction between the molecules and the matrix and (ii) to maintain proteins' stability and activity. The most common eluents used for SV separation include 0.01 to 1 M of ammonium acetate, ammonium bicarbonate and sodium acetate. Sodium chloride can be added to the buffer to improve resolution. Since the separation is based on size only in SEC, the composition of the mobile phase remains constant during the process and is known as an isocratic elution [11]. The slower the flow, the better the resolution of compounds; however, the time required for separation will significantly increase. Thus, a balance between the separation time and resolution should be made to choose a convenient flow rate between 0.1 and 1 mL/min [12]. The protein elution is typically monitored using an UV detector at 280 nm. SEC has been extensively used in SV separation, usually as a first step, for the separation of large proteins from small peptides and toxins, allowing an improved analysis of each component. Accordingly, a Superdex 200 SEC was used as a first step in the process of purification of a trimeric phospholipase A2 from *Oxyuranus scutellatus* snake venom. The protein of interest was around 45 kDa, and thus, the fraction containing molecules of 45 kDa was collected and further fractionated to obtain a pure molecule [13]. Likewise, a snake venom serine proteinase was isolated from *Vipera ammodytes ammodyets* venom using Sephacryl S-200 SEC as a first chromatographic step. Further fractionation of the peak of interest yielded 11 fractions, demonstrating the low resolution of SEC [14]. Consequently, SEC is valuable for the partition of snake venom proteins by size groups; however, it must be followed by other separation techniques, since it has an inherently low resolution and is incapable alone of isolating a single molecule from the mixture. SEC might also be used with standards to measure the MW of each fraction eluted [15].

2.1.2. Ion Exchange Chromatography

Ion exchange chromatography (IEX) is a separation technique based on the net surface charge of molecules to be separated (Figure 1(Ab)). When opting for IEX as a separation technique, multiple parameters can be optimized to match the target of the separation [16]. Accordingly, choosing a convenient stationary phase is critical for effective separation. Different types of resins can be used including cellulose, agarose, polyacrylamide, and dextran. The most used strong and weak cation columns in SV separation are sulfonic acid (SP) and carboxymethyl (CM) columns, respectively, while the most used strong anion and weak anion exchangers are quaternary amine (Q) and diethylaminoethanol (DEAE), respectively. For SV separation, the most common salts used are Tris-HCl, sodium chloride or ammonium bicarbonate. They are used for the elution of molecules mostly as linear gradients since they provide a better resolution; however, step gradient might also be used [16]. In principle, proteins constituting SVs occupy a wide pH range, and thus, both weak and strong exchangers might be used. Similarly, both cationic and anionic exchangers might be used depending on the type of protein to be isolated. If the target protein is stable at pH below its pI value (when it is positively charged), then a cation-exchange chromatography (CEX) is preferable. CEX is used at a pH lower than the pI of molecules to isolate them from the mixture, and therefore, they are used under slightly acidic mobile phase conditions with pH ranging from 5.5 to 7 [16]. Under such conditions, CEX was used to isolate three hyaluronidases from the venom of *Cerastes*. In this study, CM-sepharose CEX was used

to separate fractions previously generated by SEC. IEX provided a high resolution and an improved separation of venom components, since three different hyaluronidases were effectively purified to homogeneity using this chromatographic technique [17]. Similarly, a Mono S CEX column was used to further separate fractions generated by the SEC of *Vipera ammodytes* venom. The peak of interest was homogeneous, showing one band on SDS-PAGE and proving the high resolution of CEX as a second step of the purification process [14]. On the other hand, an SV metalloproteinase was isolated using CEX as a first step, which required a supplementary purification step to reach the pure protein [18]. In contrast, if the target protein is more stable at pH above its pI (when it is negatively charged), then an anion-exchange chromatography (AEX) is preferable. AEX is usually used with buffer pH ranging from 7 to 8.2. AEX was used as a second chromatographic step for the purification of a metalloproteinase and L-amino acid oxidase from *Bothrops atrox* and *Bothrops mojeni*. To do so, fractions eluted from SEC were re-fractionated on a Mono Q column to reach the pure protein [19]. In another study, a DEAE-Sephacel AEX column was used as a first step to separate metalloproteinase from *Bothrops moojeni* and required two successive chromatographic techniques to reach the pure protein [20]. In SV separation, IEX is widely used in both facets, knowing that SV proteins have a wide distribution across pH. In several research studies, IEX was used as a first step to separate the crude SV before evaluating its components. Mostly, it is used as a more specific intermediate step to separate a fraction originating from a previous chromatographic step.

### 2.1.3. Reversed-Phase High-Pressure Liquid Chromatography

Reversed-phase high-pressure liquid chromatography (RP-HPLC) is a high-resolution separation technique used for the partitioning of molecules with different polarities (Figure 1(Ac)). Due to its high resolution, RP-HPLC is used in almost every SV separation, since it provides a fair separation of protein isoforms that are very abundant in these mixtures [21]. Numerous columns are developed and are commercially available constituting of hydrophobic alkyl groups usually attached to silica beads. The most common columns are the butylsilane (C4), octylsilane (C8), and octadecylsilane (C18), which are made of 4-, 8- or 18-carbon chains, respectively. These columns differ in their hydrophobicity: the longer the carbon chain, the more hydrophobic the column, and therefore the stronger the interaction with the sample, causing an increased retention and better separation. Consequently, C18 is the most employed column for the separation of complex mixtures such as SVs [22]. The use of C4 and C8 columns has also been noted in SV separation. Molecules bound with different strengths to the matrix are then eluted using organic solvents such as propanol, methanol and acetonitrile [23]. The latter is the most used for SV separation. The application of a gradient organic solvent allows the elution of molecules in order of increasing hydrophobicity, and the gradients used could be linear or step gradients, depending on the features of the molecules to be isolated. RP-HPLC is very versatile and can be employed for several objectives. This technique was thoroughly used for the isolation and purification of SV molecules. For this purpose, RP-HPLC is usually coupled with other separation techniques and used as the last step for better purification. C18-based RP-HPLC was used to isolate a CRiSP from *Crotalus oreganus helleri* venom. Crude venom was fractionated in 27 fractions with F7 having molecules of 25 KDa (MW range of CRiSP family) and thus was further chromatographed using CEX. The chromatographic profile showed one major peak that was identified as CRiSP and a minor peak demonstrating the high resolution of C18 columns [24]. A C4 column was used as a last step to purify a metalloproteinase from *Bothrops pauloensis* venom after IEX and SEC. C4 fractionation resulted in a highly pure protein [15]. C4 was also used for the analysis of *Peruvian pit vipers*' whole venom. The SDS-PAGE of resulting fractions was extremely heterogeneous, indicating the need for another separation technique [25]. Another usage of RP-HPLC is to confirm the purity of isolated proteins. Accordingly, the purity of hyaluronidase isolated from *Crotalus durissus terrificus* venom using three successive chromatographic steps (CEX, SEC and hydrophobic chromatography) was confirmed

using a C18 column showing a single peak on chromatogram [26]. RP-HPLC might also be used for the analysis of whole SV composition or the comparison of SVs variability [27,28]. Most importantly, RP-HPLC is the most compatible chromatographic method to be used for the downstream identification of SV proteins with mass spectrometry.

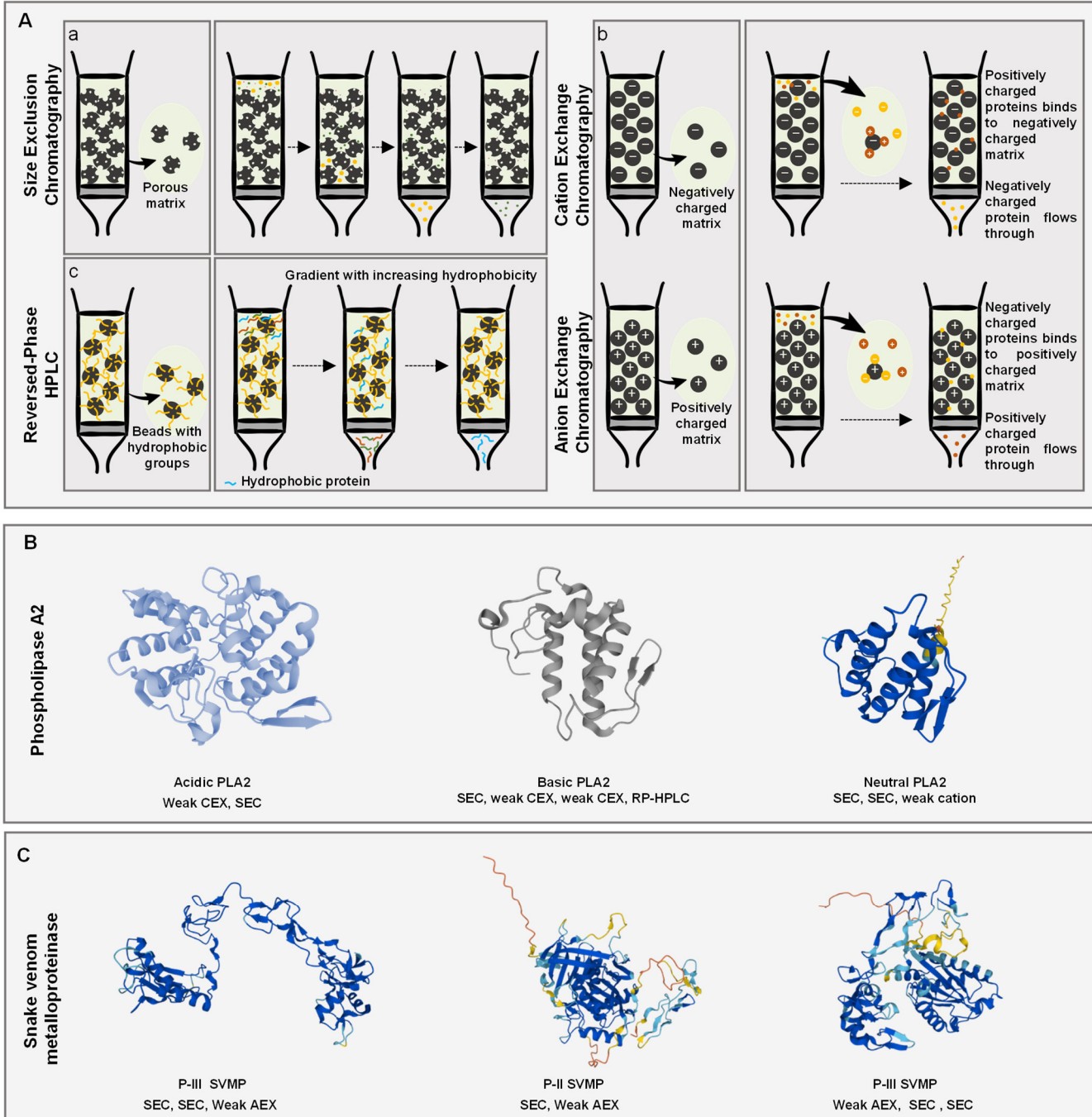

**Figure 1.** (**A**) Schematic representation of the following chromatographic techniques. (**Aa**) Porous matrix of size exclusion chromatography (i.e., dextran, dextran–agarose, or dextran–polyacrylamide) and the separation process using an isocratic elution buffer (i.e., 0.01 to 0.1 M of ammonium acetate, ammonium bicarbonate or sodium acetate). (**Ab**) Anion-exchange chromatography (AEX) positively

charged matrix (i.e., quaternary amine or diethylaminoethanol) and cation-exchange chromatography (CEX), negatively charged matrix (i.e., sulfonic acid or carboxymethyl) and their corresponding separation process using linear or stepwise elution gradients (i.e., sodium chloride or ammonium bicarbonate can be used as elution buffers). (**Ac**) Hydrophobic matrix of reversed-phase high-pressure liquid chromatography (i.e., silica matrix beads with varied length alkyl groups and the separation process in RP-HPLC using linear or stepwise gradients of acetonitrile. (**B**) Examples of three phospholipase A2 (acidic, basic, and neutral PLA2) isolated using the aforementioned chromatographic techniques. From the left to the right an acidic PLA2 from *Daboia siamensis* [29], a basic PLA2 from *Protobothrops flavoviridis* [30] and a neutral PLA2 from *Naja naja sputatrix* [31] (**C**) Snake venom metalloproteinases (PIII and PII SVMP) isolated using the aforementioned chromatographic techniques. From the left to the right a PIII SVMP from *Bothrops leucurus* [32], a PII SVMP from *Gloydius brevicaudus* [33], and a PIII SVMP from *Trimeresurus gramineus* [34]. ((**B**): acidic PLA2 UniProt: P31100, basic PLA2 UniProt: P0DJJ9, neutral PLA2 UniProt: Q92084; (**C**) PIII SVMP UniProt: P86092, PII SVMP UniProt: Q9YI19, PIII SVMP Uniprot: P0C6E8).

### 2.1.4. Affinity Chromatography

Affinity chromatography is a separation technique used to isolate a specific molecule or group of similar molecules from a complex mixture (Figure 2). This type of chromatography is considerably deployed for the separation and purification of a variety of SV proteins, since it provides a less costly and less time-consuming method [35]. Several molecules have been developed and customized as substrates to bind specific SV proteins from the crude mixture. For instance, an SV metalloproteinase having a strong interaction with ssp-3 protein was isolated by immobilizing ssp-3 on a matrix [36]. The substrate of PLA2 enzyme was synthesized and immobilized on an affinity column for the purification of the protein from *Crotalus durissus* venom [37]. Likewise, PLA2 was isolated from *Bothrops jararaca* venom using immunoaffinity chromatography. For this process, an anti-crotoxin IgG, with high affinity to PLA2, was developed and immobilized on Sepharose polymer, allowing the purification of the protein [38]. Alternatively, commercially available affinity columns can be used along with other chromatographic steps to separate a specific group of proteins. Since these columns are not specific to a single molecule, a series of chromatographic steps are used to reach protein purity. Three types of columns are commonly used for the separation of SVs bearing lactose, heparin and benzamidine substrates. Columns with immobilized lactose were used to isolate C-type lectins from *Bothrops atrox* and *Cerastes cerastes* venoms, resulting in a highly pure protein [39,40]. Similarly, these columns can be used to generate a lectin-free sample to further isolate molecules from the eluate [41]. Heparin columns are used to isolate coagulation factors having affinity for heparin, while benzamidine columns are preferably used to isolate serine proteases. For example, a heparin affinity column was used to isolate a metalloproteinase from *Bothrops moojeni* venom after two chromatographic steps: IEX and SEC [20]. On the other hand, serine protease was isolated from *Bothrops pirajai* venom using benzamidine columns along with size exclusion and reverse phase chromatography [42]. Other serine proteinases were isolated from *Bothrops atrox* and *Bothrops brazili* venoms using SEC and benzamidine affinity chromatography [43]. To elute molecules from affinity columns, different types of buffers can be used such as decreasing the pH step gradient, salt gradient or introduction of a competitor molecule for the binding sites. Affinity chromatography requires in some cases the use of strongly acidic buffers (pH = 2.5) to disrupt interactions between the molecule and the stationary phase. This might interfere with the protein structure and function; thus, appropriate buffers should be selected depending on the objective of the separation. Another limitation to be aware of is the non-specific binding of proteins that would contaminate the eluted protein [35].

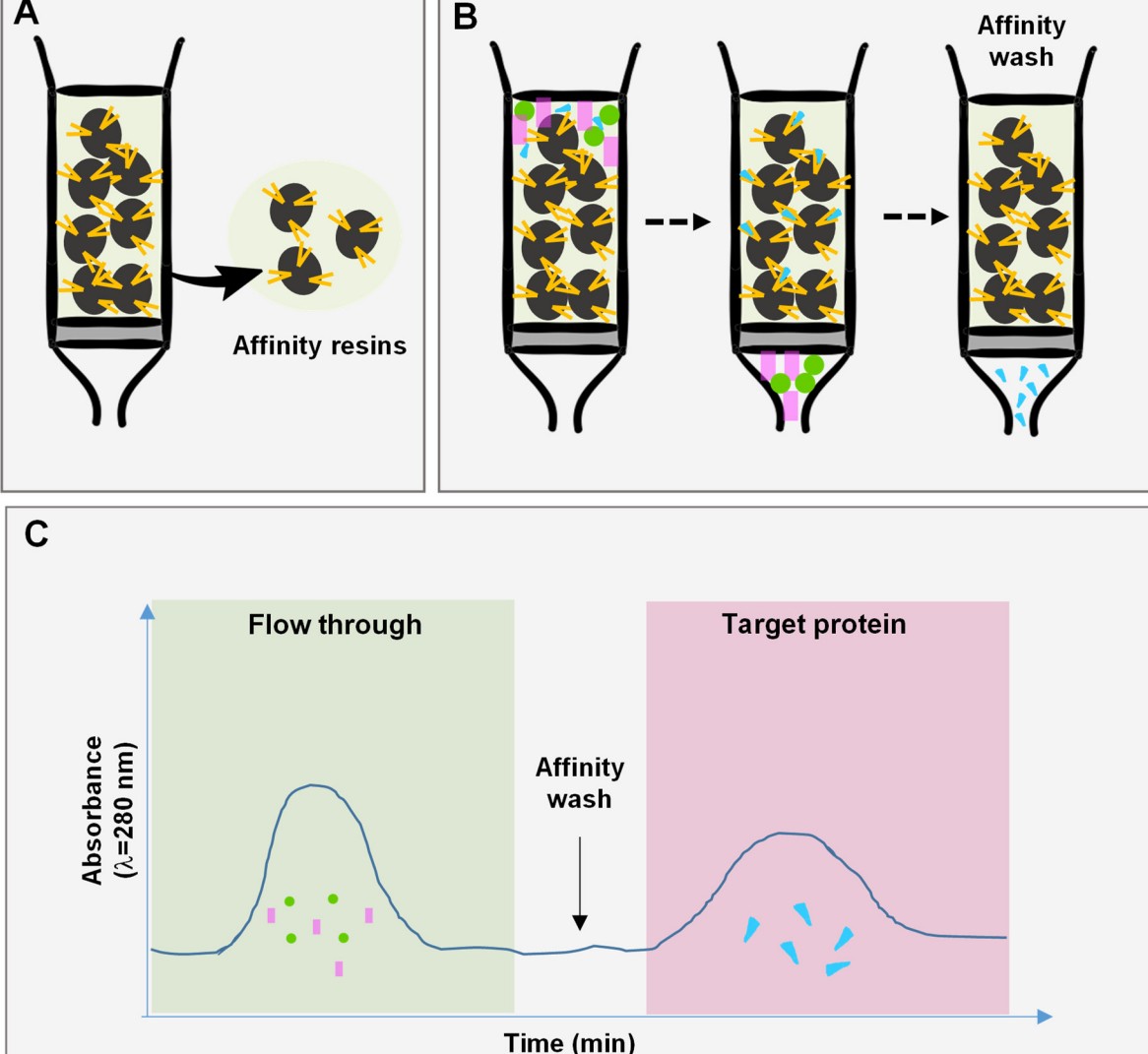

**Figure 2.** Schematic representation of affinity chromatography. (**A**) The affinity resins (i.e., agarose) that binds the molecule of interest and (**B**) the separation process, which includes an equilibration step (using Tris-HCl or sodium acetate) followed by the elution (using Tris-HCL/Glycine-HCl, PBS). (**C**) A typical chromatogram of affinity chromatography with the first flow through containing only molecules with no affinity to the matrix, which is followed by affinity wash to disrupt the interaction between the matrix and the target protein and elute the latter.

## 2.2. Electrophoretic Techniques

The high versatility, feasibility, and practicality of chromatographic techniques outweighed those of electrophoretic techniques; nevertheless, the latter remain an important means for the separation of a variety of venoms (Table 1). For SVs, electrophoretic techniques are used rigorously for the separation and identification of protein content of whole venom and fractions. Even though electrophoretic methods have several limitations, their use is integral in the process of SV analysis. Two major types of electrophoretic methods are currently used: one-dimensional gel electrophoresis and two-dimensional gel electrophoresis.

**Table 1.** The advantages and disadvantages of different separation techniques used for snake venom analysis.

| Separation Technique | Advantages | Disadvantages |
|---|---|---|
| **Size exclusion chromatography** | - Simple to design (isocratic gradient)<br>- Arranges SV proteins into fractions of known MWs<br>- No sample loss since solutes do not interact with stationary phase<br>- Short analysis time | - Low resolution compared to other techniques<br>- Limited number of peaks can be resolved<br>- Sample evidently requires further separation to reach purity |
| **Ion-exchange chromatography** | - Resins have long life<br>- Buffers are suitable and non-denaturing<br>- High-resolution separation | - Can only be used for charged molecules<br>- Small changes in pH alter the binding of molecules |
| **Affinity chromatography** | - High specificity<br>- High resolution<br>- Purification could be performed in a single step | - Harsh buffers used could alter the structure and function of target proteins<br>- Resin development is time-consuming and costly |
| **RP-HPLC** | - Rapid, efficient, and accurate<br>- Use of water-based solvents that are compatible with SV proteins<br>- Most compatible with MS | - Costly organic solvents<br>- Coelution of molecules with same polarity |
| **1D gel electrophoresis** | - Gives an idea of the distribution of molecules in each fraction<br>- Helps determine protein molecular weights<br>- Requires small amount of sample | - Procedure is time consuming<br>- Large amounts of sample could not be separated at once<br>- Suitable only for whole proteome analysis |
| **2-D gel electrophoresis** | - Good for comparison of snake venoms<br>- Whole proteome analysis could be performed in a single separation step | - Inefficient separation<br>- Poor reproducibility<br>- Spot trains (hard to analyze each spot apart)<br>- Low amount of proteins prevents identification my MS |

### 2.2.1. One-Dimensional Gel Electrophoresis (1-DGE)

SDS-PAGE is considerably used in the process of SV separation for distinct objectives, including proteomic analysis. In fact, SVs are separated by SDS-PAGE followed by in-gel protein digestion and mass spectrometry for protein identification [44]. A chromatography step might also be added before gel separation to improve resolution and enhance protein coverage [14]. For example, HPLC fractions of *Crotalus durissus* crude venom were separated using SDS-PAGE under reducing conditions; bands were excised, digested, and subjected to mass spectrometry for protein identification [45]. In addition to proteomic analysis, SDS-PAGE is frequently used during the multidimensional chromatography of SVs. It is useful as an intermediate step to determine the protein content of each fraction and the target fractions to be further separated. It might be used after an initial chromatographic step to evaluate the distribution of proteins [46], as it might be used as the last step to confirm the purity and molecular weight of the isolated protein. An example is the use of SDS-PAGE to analyze a fraction from RP-HPLC, which showed a single band with a molecular weight corresponding to PLA2, indicating the purity of the isolated molecule [47].

However, mass spectrometry (MS) identification of the molecular weight remains more accurate. SDS-PAGE might be used to unravel the intra- and inter-protein interactions. A reducing agent is added to the sample to reduce disulfide bonds and allow each protein subunit to migrate independently in the gel [48]. The process can therefore be conducted under both reducing and non-reducing conditions to compare protein migration under different conditions [49].

2.2.2. Two-Dimensional Gel Electrophoresis (2-DGE)

This technique is also significantly used in SV analysis for several purposes. In general, 2-DGE is useful to unravel the overall distribution pattern of venom proteins and obtain information regarding their pI and MW distribution [50]. It is fairly used for the identification of protein families constituting the venom. It was therefore used to perform the proteomic characterization of several SVs such as coral snake *Micrurus pyrrhocryptus* [51], *Macrovipera lebetina obtusa* [52], and *Bothrops insularis* [53]. The procedure used to analyze these venoms involves the excision of proteins spots from the gel, trypsin digestion followed by mass spectrometry analysis for protein identification. Additionally, 2-DGE could be used coupled with other chromatographic steps to isolate proteins from SV mixtures. 2-DGE was used after affinity chromatography to evaluate the flow through of the column for separation of several SVs [54]. PLA2 was isolated from *Bothrops mattogrossensis* venom using two chromatographic steps; then, 2-DGE was performed to determine the MW (13.55 kDa) and pI (9.5) of the isolated molecule [55] Moreover, 2-DGE could be used for the comparison of SV proteomes. Venoms of same species could be compared for snakes living under different conditions; otherwise, venoms from different species could be compared to obtain a better understanding of the similarities and differences among species. For example, Wongtay et al. used 2-DGE to evaluate the differences in SV composition of three different *Ophiophagus hannah* snakes originating from different locations in Thailand. Results showed differences in protein migration in the gels and in protein spot identification [56]. Lastly, 2-DGE plays an important role in identifying the selectivity of antivenoms, since proteins from the gel could be transferred onto a membrane for immunoblotting analysis using antivenoms. This technique was used to identify the reactivity of a polyclonal antivenom against proteins of Egyptian cobras and indicated a weak immunoreactivity toward low molecular weight proteins, suggesting the need for the further development of more specialized antivenoms for these species [57].

Table 1 summarizes the main advantages and disadvantages of chromatographic and electrophoretic techniques used in SV analysis.

## 3. Implementation of Separation Methods for SVs

In the following part, we will detail the implementation of separation strategies used for SV analysis. Appropriately, different separation techniques are used to meet the objectives of the study that could be to (i) isolate and purify a single component or to (ii) analyze whole venom proteome.

### 3.1. Bioassay-Guided Fractionation

It uses biological assays to perform a fractionation aiming to isolate a specific molecule. During the process, a complex mixture is separated typically by chromatographic techniques, and each fraction is tested separately for a specific biological feature. The fractions of interest are then re-fractionated until a pure protein is recovered (Figure 3) [58]. The choice of the assay to be used is directly related to the molecule to be isolated. For SVs, numerous bioassays are currently used for different specific components. In fact, SVs induce a wide spectrum of effects once injected in prey, since they target several organ systems: namely, the nervous and the cardiovascular systems [59]. Different bioassays are developed for the isolation of cardiotoxic and neurotoxic components. Since SV cardiotoxic components are known to have effects on the cardiac muscle or vascular smooth muscle [59], several bioassays might be used for the isolation of cardiotoxic components.

Firstly, Langendorff model heart preparations could be used to screen for cardiotoxic molecules. For example, this model was used to screen for the cardiotoxic component of *Vipera ammodytes* venom. First, the venom was fractionated using SEC and generated four fractions with fraction C having the highest cardiotoxicity. The subfraction C1 induced an irreversible cardiac arrest of the isolated heart, while subfraction C2 induced an irreversible significant decrease in heart rate without inducing cardiac arrest. Consequently, both fractions were re-fractionated using an RP-HPLC C4 column, and Ammodytin L (AtnL) was identified as the major protein inducing cardiotoxicity [60]. Another robust biological model to analyze cardiotoxicity could be the zebrafish model. This model is used to monitor atrial and ventricular rates, blood flow and clot formation. Recently, we took advantage of zebrafish embryos transparency to assess the direct cardiotoxicity of SV [61]. In addition, the zebrafish model was used to isolate the cardiotoxic molecule from *Lachesis muta* venom. First, crude venom was fractionated using SEC; then, the active fractions were fractionated using AEX, and fraction 8 with the highest cardiotoxicity was further purified using an RP-HPLC C18 column, generating a pure protein "mutacytin-1" [62]. Moreover, isolated rat mesenteric arteries might also be used to evaluate the vasorelaxant effect of fractions. This technique was used to isolate the vasorelaxant molecule in *Bothrops leucurus* venom. Crude venom was fractionated with CEX followed by RP-HPLC for the vasorelaxant fraction. After MS analysis, the molecule inducing vasorelaxation turned to be a PLA2 [63]. On the other hand, several neurotoxic components have been isolated and identified from SVs by the means of bioassay-guided fractionation. One of the most common neurotoxic assays used is the chick biventer cervicis nerve-muscle preparation. Neurotoxic components usually induce an inhibition of indirect twitches, leading to a decrease in muscle contraction in a concentration-dependent manner. This assay was used in tandem with SEC to isolate hostoxin-1 from *Hoplocephalus stephensi* [64], rufoxin from *Rhamphiophis oxyrhynchus* [65], and SPAN from *Austrelaps* species [66]. Another method to test neurotoxicity is the evaluation of neurotoxic effects on the sciatic nerve. The neurotoxic component of the venom of *Daboia russelii* was identified using this bioassay. The venom was fractionated by SEC and fraction 13 induced neurotoxic symptoms, such as respiratory distress, hind limb paralysis, lacrimation, convulsions, and profuse urination. This fraction was further purified using RP-HPLC, and the isolated protein was shown to inhibit indirectly stimulated twitches of sciatic nerve–muscle preparations [67]. Of note, SDS-PAGE is very useful in bioassay-guided fractionation, since it can indicate the protein content of each fraction eluted and the purity of the isolated protein [61]. Different combinations of separation techniques have been used for the isolation of novel SV molecules that upon biological characterization implicate the clinical outcome of the toxins or their potential use for drug development. Table 2 summarizes the most recent techniques used for the purification of the most common SV protein families.

**Table 2.** Different separation strategies used most recently to isolate the most abundant SV protein families. Da: Dalton, ND: Not Determined, [a] MW determined by SDS-PAGE, [b] MW determined by MS.

| Molecule | SV | M.W. | Strategy of Separation | Columns Used | Reference |
|---|---|---|---|---|---|
| | *Crotalus molossus nigrescens* | 13,972 Da [b] | RP-HPLC | C18 | [68] |
| | *Naja sumatrana* | 15,606 Da [b] | Size exclusion Size exclusion RP-HPLC | Sephadex G-50 Superdex 75 10/30 GL C18 | [69] |
| Phospholipase A2 | *Daboia siamensis* | 14,000 Da [a] | Weak cation exchange Size exclusion | CM-FF Superdex 75 10/300 GL | [29] |
| | *Bothrops atrox* | 13,826 Da [b] | Weak cation exchange RP-HPLC | CM-sephadex C-25 C18 | [70] |
| | *Micrurus lemniscatus* | 13,568 Da [b] | RP-HPLC | C8 | [71] |

**Table 2.** *Cont.*

| Molecule | SV | M.W. | Strategy of Separation | Columns Used | Reference |
|---|---|---|---|---|---|
| **Metalloproteinase** | *Daboia siamensis* | 68,000 Da [a] | Size exclusion<br>Strong Anion exchange<br>Strong cation exchange | Superdex 75 10/300 GL<br>Mono Q<br>Resource S | [29] |
| | *Bothrops atrox* | 25,000 Da [a] | Size exclusion<br>Strong anion exchange | Superdex 200<br>Mono Q 5/50 GL | [19] |
| | *Cerastes cerastes* | 35,000 Da [a] | Size exclusion<br>Weak anion exchange<br>Affinity | Sephadex G-75<br>DEAE sephadex A-50<br>Benzamidine Sepharose 6B | [72] |
| | *Bothrops moojeni* | 25,000 Da [a] | Weak cation exchange<br>RP-HPLC | CM-FF<br>C18 | [18] |
| | *Vipera ammodytes* | 21,000 Da [a] | Size exclusion<br>Strong cation exchange | Superdex 75 10/300 GL<br>SP sepharose | [73] |
| **L-amino acid oxidase** | *Bothrops moojeni* | 58,000 Da [a] | Size exclusion<br>Strong anion exchange | Superdex 200<br>Mono Q 5/50 GL | [19] |
| | | ND | Weak cation exchange<br>Hydrophobic interaction<br>Affinity | CM-Sepharose<br>Phenyl-Sepharose CL-4B<br>Benzamidine Sepharose | [74] |
| | *Cerastes cerastes* | 58,000 Da [a] | Size exclusion<br>Strong anion exchange<br>Affinity | Sephadex G-75<br>Resource Q<br>HiTrap heparin | [75] |
| | *Cerastes viper* | 60,000 Da [a] | Size exclusion<br>Weak anion exchange | Sephacryl S-200<br>DEAE-Sepharose | [76] |
| | *Micrurus mipartitus* | 57,000 Da [a] | Size exclusion<br>RP-HPLC | Biosec S-200<br>C18 | [77] |
| **Serine protease** | *Bothrops jararaca* | 28,000 Da [b] | Size exclusion<br>Weak anion exchange<br>RP-HPLC | Sephacryl 200<br>DEAE<br>C18 | [78] |
| | *Crotalus simus* | 24,600 Da [a]<br>31,300 Da [a] | Size exclusion<br>Size exclusion<br>RP-HPLC | Sephadex G-200<br>Sephadex G-75<br>C5 | [79] |
| | *Crotalus durissus collilineatus* | 29,474 Da [b]<br>28,388 Da [b] | Size exclusion<br>Strong anion exchange<br>RP-FPLC | Sephacryl S100 HR<br>Mono Q 5/50GL<br>C4 | [80] |
| | *Bothrops moojeni* | 30,300 Da [a] | Weak cation exchange<br>RP-HPLC | CM Sepharose<br>C18 | [81] |
| **C-type lectin** | *Bothrops alternatus* | 25,000 Da [a] | Weak anion exchange<br>Affinity<br>RP-HPLC | DEAE-Sephacel<br>HiTrap heparin HP<br>RP-source 15 RPC<br>ST4.6/100 | [82] |
| | *Lachesis muta muta* | 28,000 Da [a] | Size exclusion<br>Strong anion exchange<br>RP-HPLC | Sephacryl 300<br>Mono Q2<br>C18 | [62] |
| | *Micrurus surinamensis* | 23,461 Da [b] | Size exclusion<br>Weak anion exchange<br>Size exclusion | Sephacryl S-200<br>DEAE-Sepharose<br>Superdex G75 10/30 | [83] |
| | *Macrovipera lebetina* | ND | Size exclusion<br>Strong cation exchange | Sephadex G-75<br>Mono S | [84] |
| | *Bothrops jararacussu* | ND | Affinity<br>RP-HPLC | Sepharose 6B-CL-lactose<br>C18 | [85] |
| | *Cerastes cerastes* | 34,271 Da [b] | Affinity<br>Size exclusion<br>RP-HPLC | Sepharose 4B-lactose<br>Sephadex G-25<br>C8 | [39] |

**Table 2.** *Cont.*

| Molecule | SV | M.W. | Strategy of Separation | Columns Used | Reference |
|---|---|---|---|---|---|
| **Cysteine-Rich Secretory Protein** | *Naja kaouthia* | 24,900 Da [b] | RP-HPLC<br>RP-HPLC | C18 (5 μm)<br>C18 (3 μm) | [86] |
| | *Crotalus oreganus helleri* | 25,000 Da [a] | RP-HPLC<br>Strong cation exchange | C18<br>SP 5 PW | [24] |
| | *Bothrops alternatus* | 24,400 Da [a] | Weak anion exchange<br>Size exclusion<br>Affinity | DEAE-Sepharose<br>Sephacryl S-100<br>Affi-gel blue<br>Sepharose | [87] |
| | *Bothrops jararaca* | 24,600 Da [a] | Size exclusion<br>Strong anion exchange<br>RP-HPLC | Sephacryl S-200<br>15 Q<br>C18 | [88] |
| **Disintegrins** | *Crotalus totonacus* | 7437 Da [b] | Size exclusion<br>RP-HPLC | Sephadex G-75<br>C18 | [89] |
| | *Crotalus durissus collilineatus* | 7287 Da [b] | RP-FPLC<br>RP-FPLC | C18 (5 μm)<br>C18 (3.6 μm) | [90] |
| | *Cerastes cerastes* | 7083 Da [b] | Size exclusion<br>Weak anion exchange<br>RP-HPLC | Sephadex G75<br>DEAE-Sephadex A50<br>C8 | [91] |
| | | 13,835 Da [b] | Affinity<br>Size exclusion | Sepharose 4B-lactose<br>Sephadex G-50 | [45] |
| | *Vipera ursinii* | 14,018 Da [b] | Size exclusion<br>RP-HPLC | Sephadex 75 10/300<br>GL<br>C18 | [92] |
| **Three-Finger Toxins** | *Naja nigricollis* | 6743 Da [b] | RP-HPLC<br>RP-HPLC | C18<br>C4 | [93] |
| | *Micrurus tschudii* | 6538 Da [b] | RP-HPLC | C18 | [94] |
| | *Naja melanoleuca* | 7441 Da [b]<br>7756 Da [b]<br>7787 Da [b]<br>8030 Da [b] | Size exclusion<br>Weak cation exchange<br>RP-HPLC | Sephadex G50<br>Bio 1000 CM<br>C18 | [95] |

### 3.2. Whole Proteome Characterization and Identification

The currently available separation and analytical techniques allowed researchers to isolate and identify the most abundant constituents of the venoms. However, given the complexity of SVs, numerous molecules remain unexplored. More recently, the fundamental analysis of complex SV proteomes was made less challenging owing to the evolution of proteomics field. Accordingly, several proteomic techniques have been adapted, providing a rapid and relatively inexpensive method for the decomplexation of venom mixtures [96–98]. Both proteomic approaches, bottom–up (BU) and top–down (TD), might be used to analyze SVs composition. The latter has not been used in snake venomics until recently, since it is still in the course of progression [98,99]. On the other hand, bottom–up proteomics is more frequently used and typically involves the identification of trypsin-digested proteins by tandem mass spectrometry (MS/MS) after separation by both chromatographic and/or electrophoretic techniques. Multiple workflows are used for the BU proteomic analysis of SVs (Figure 4).

The first workflow (Figure 4A) involves the separation of venom mixture with RP-HPLC using a stepwise or linear acetonitrile gradient. This step is followed by an in-solution trypsin digestion of fractions and LC-MS/MS. Prior to digestion, collected fractions can be tested using SDS-PAGE to evaluate the distribution of proteins in each fraction regarding the MW and number of protein bands. This technique was extensively used in the venomics field to reveal the proteomic composition of SVs such as *Bungarus sindanus*, *Calliophis intestinalis*, *Deinagkistrodon acutus*, *Trimeresurus wiroti*, *Trimeresurus puniceus*, and *Hydrophis curtus* [100–105]. This technique was also used in several studies to compare the variability between different SVs [106].

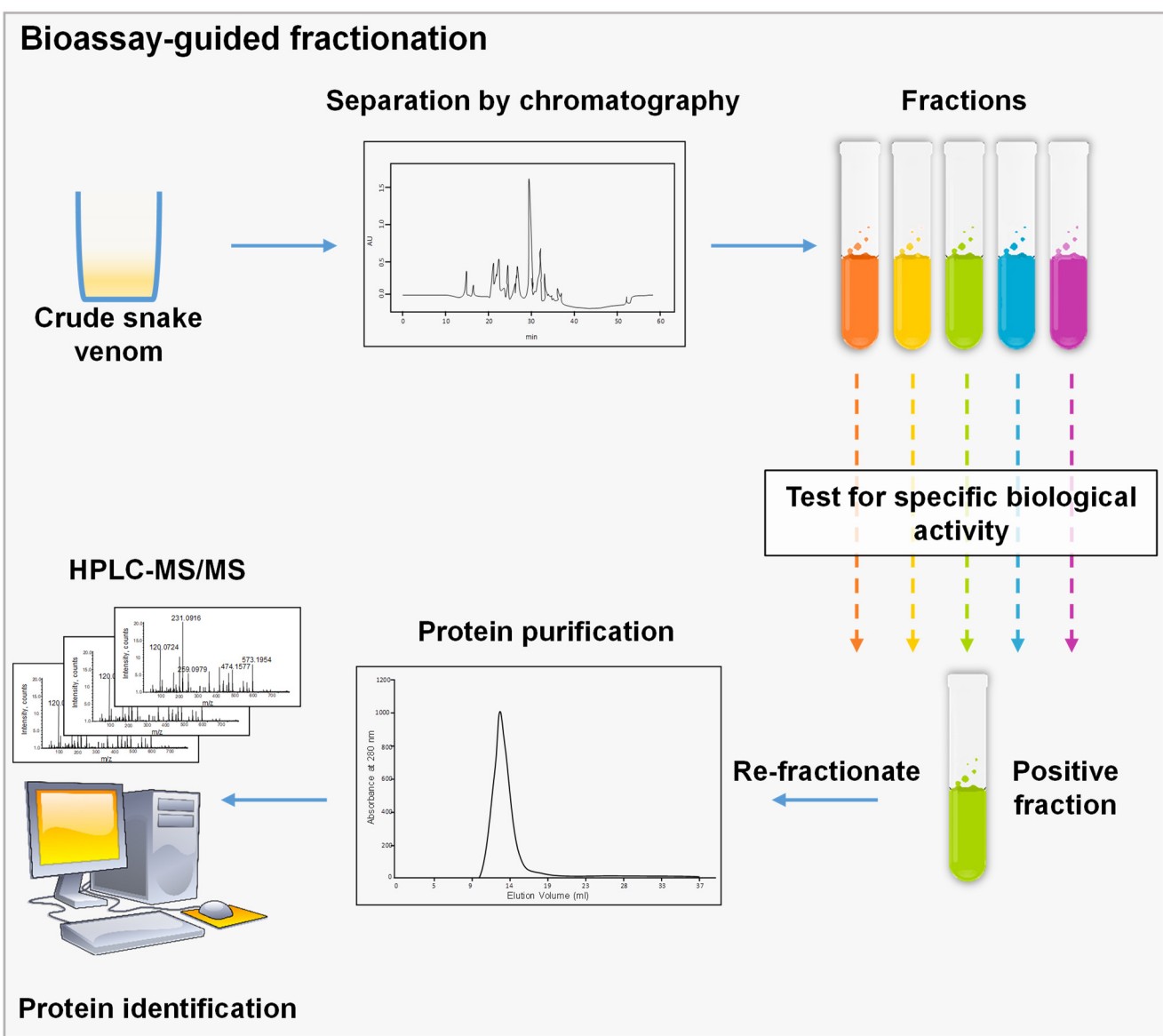

**Figure 3.** Schematic representation of bioassay-guided fractionation, crude venom is separated using a chromatographic technique; then, fractions are assessed using a specified biological assay to choose the corresponding fraction. This step could be repeated several times until we reach a single peak or a pure molecule that could be identified by MS.

The second workflow (Figure 4B) involves the use of a electrophoretic separation. In this approach, crude venom is separated by SDS-PAGE, and protein bands are visualized using Coomassie Blue or silver stain. Later, the protein bands are cut, digested, and then analyzed by LC-MS/MS. An RP-HPLC step might be added before the gel separation to increase the resolution of the workflow. This approach was considerably used in venomics for distinct SVs. For example, it was used to characterize and compare venoms of *Trimeresurus macrops* and *Trimeresurus hageni* and led to the identification of 187 and 216 proteins, respectively [107].

In the third workflow (Figure 4C), 2-DGE is first performed and followed by the in-gel digestion of all protein spots that are then analyzed by LC-MS/MS. The strategy was shown to be effective for the separation and identification of several SV proteomes [108,109]. As with every technique, 2-DGE has its own advantages and disadvantages. In fact, it provides the advantage of high molecular weight proteins recovery; however, it is sometimes hard to distinguish protein spots due to inaccurate separation and spot trains [110]. In

addition, if protein quantity is insufficient in protein spots, then further analysis might not be performed [111]. This technique has been used to analyze several snake proteomes such as *Gloydius intermedius*, *Agkistrodon contortrix* and *Naja asehi* [108–110]. A study comparing different workflows indicated that the in-gel digestion of proteins provides a lower resolution compared with direct in-solution digestion [112]. However, after HPLC, in-gel protein digestion provides higher resolution compared to in-solution protein digestion [113]. These findings show that the choice of the method should be related to the properties of the proteins present in the mixture and to the overall workflow adopted for the analysis.

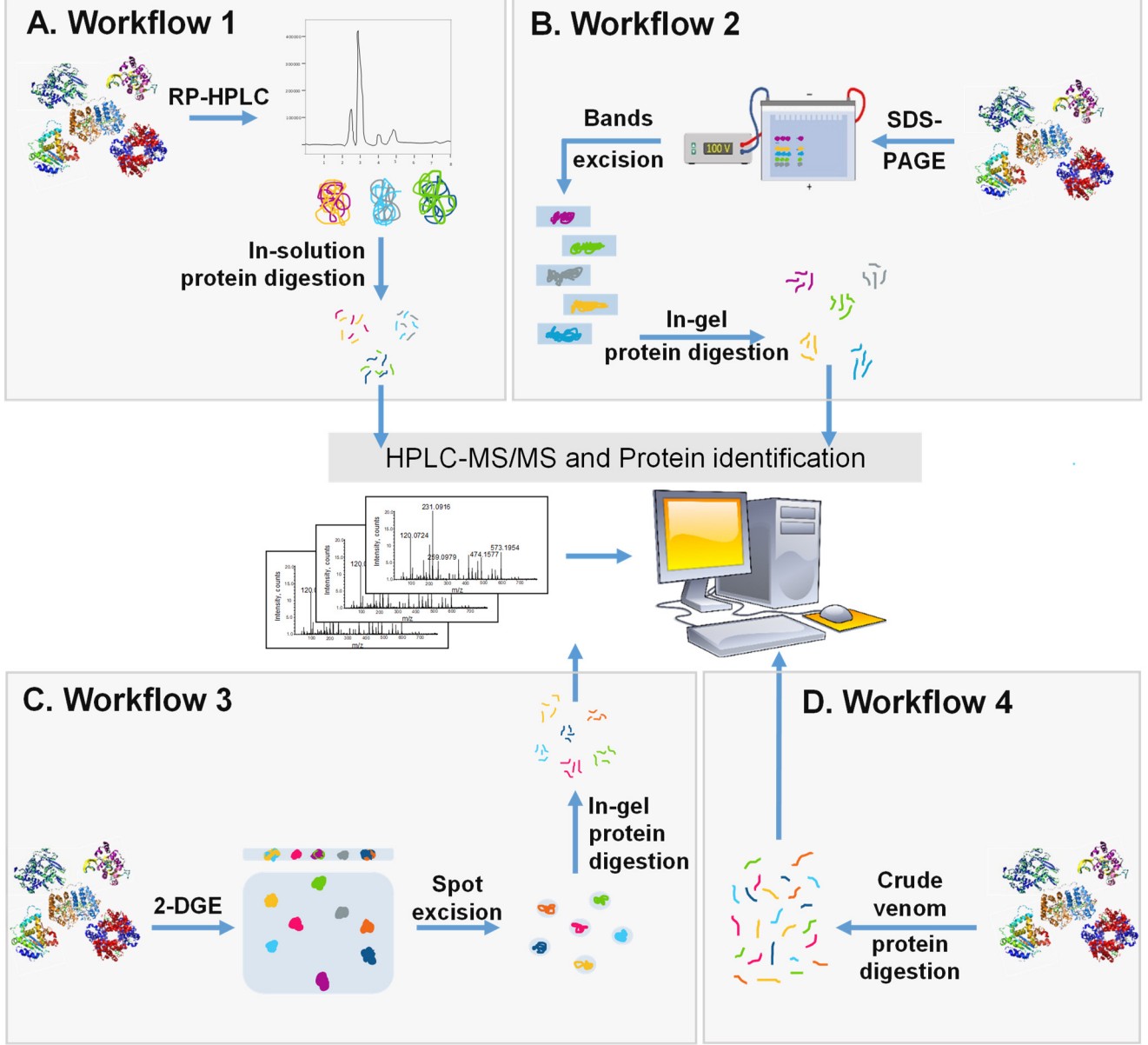

**Figure 4.** Schematic representation of the 4 major workflows used for whole venom separation and proteome identification. (**A**) Crude SV is separated with RP-HPLC followed by in-solution trypsin digestion and the identification of components by LC-MS/MS. (**B**) Crude SV is separated by SDS-PAGE and protein bands are cut, digested, and analyzed using LC-MS/MS. (**C**) Crude venom is separated using 2-DGE followed by protein spot excision, in-gel digestion, and then protein identification by LC-MS/MS. (**D**) Crude venom is directly digested into peptides and analyzed using LC-MS/MS.

The fourth workflow (Figure 4D) is known as shotgun analysis. This technique involves the tryptic digestion of crude venom proteins. Ultimately, the resulting peptides are separated by RP-HPLC using a C18 column followed by MS to identify proteins. Shotgun analysis has numerous advantages over other techniques, since it does not require a decomplexation process at the beginning that is time consuming and, in some cases, expensive. This strategy allows the recovery of low abundant proteins, since they may be lost during the separation step performed in other techniques [114]. This is reflected with a relatively high number of protein hits compared to other techniques. For example, the analysis of *Vipera ammodytes ammodytes* venom with the shotgun technique identified 99 proteins that is folds higher than other workflows used to analyze this SV [115]. This technique was also used for the analysis of *Deinagkistrodon acutus* venom and led to the identification of 84 proteins [116]. Despite the numerous advantages, some limitations have been reported such as the challenging identification of low molecular weight protein [115]. One way to overcome this limitation is to employ a combinatorial peptide ligand library and membrane filtration to recuperate low MW proteins [117]. This allows concentrating low-abundance proteins in the mixture, leading to a better protein coverage by shotgun analysis. The analysis of *Naja ashei* venom by Hus et al. indicated that the shotgun technique uncovers proteins that are not identified by other techniques such as 2-DGE, in which 19 proteins were identified compared to 39 proteins identified with shotgun. In this study, the authors indicate that a decomplexation step, although being time and labor consuming, provides higher coverage of proteins [110]. In another study, *Bothrops leucurus* venom yielded 137 proteins hits for shotgun analysis compared to 62 proteins identified using the first workflow [114].

Ultimately, even with the great advancement in the proteomic field, there is still no standard technique for SV analysis. Rather, a combination of several techniques could be used depending on the available resources and objectives of the study. This has been shown to provide a better coverage of the proteomes, since each technique might identify different sets of proteins. Proteomes of *Naja mossambica* and *Naja naja nigricincta* were analyzed using three different approaches, and the merged data identified 75 and 73 proteins, respectively, which was higher than the proteins identified by each approach alone [106]. Similarly, *Naja naja nigricincta* and *Bungarus caeruleus* venoms were analyzed by three different approaches simultaneously, using the second and fourth workflows, identifying a total of 81 and 46 proteins, respectively, for each SV [118].

To date, the literature available on snake venomics indicates undoubtedly that bottom–up proteomics is the strategy of choice for venom proteomes analysis. Yet, the use of such a strategy for the analysis of complex mixtures such as SVs is particularly challenging. This is due, in general, to the protein digestion step required prior to MS analysis. TD analysis provides an alternative to solve this problem, since intact proteins might be analyzed directly [119]. TD was not used in the field of venomics until a few years ago, and its use is limited, since it requires very sophisticated and costly instrumentation. As with any other type of analysis, TD requires the separation of complex protein mixtures prior to MS. Therefore, RP-HPLC constitutes the best method for separation, since it is compatible with the top–down downstream analysis [120]. However, other separation methods might be used as well [121]. A multidimensional separation seems to provide the best resolution, and it includes the use of gel-based separations followed by RP-HPLC [121]. TD proteomics was used successfully for the analysis of several SV proteomes including *Dendroaspis jamesoni*, *Dendroaspis kaimosae*, *Dendroaspis viridis*, and *Ophiophagus hannah* venoms [122,123]. In addition to being costly, TD analysis has other limitations, including its inconvenience for the identification of high molecular weight proteins that are an abundant constituent of viperid snakes [119,121]. Thus, to overcome the limitations of both strategies, an integration of TD and BU proteomics is currently employed and is designated as 'middle–down proteomics'. This approach was used for the analysis of *Echis carinatus sochureki*, *Protobothrops flavoviridis*, and *Vipera anatolica senliki* venoms and proved to be the most effective compared to each method alone [124,125]. In Table 3, we summarize the

different workflows and strategies detailed previously with corresponding examples of analyzed SVs and consequent findings.

**Table 3.** Different workflows and strategies used in SV proteomic analysis, applications, and consequent findings. Da: Dalton.

| Workflow | SV | Nb of Protein Families | Nb of Proteins | Main Protein Distribution/Most Abundant Venom Components | Reference |
|---|---|---|---|---|---|
| **Workflow 1 RP-HPLC/in-solution trypsin digestion/LC-MSMS** | *Trimeresus wiroti* | 10 | 62 | SV serine protease (31.04%) <br> SV metalloproteinase (26.17%) <br> Disintegrins (9.08%) <br> C-type lectins/snaclecs (8.05%) <br> Phospholipase A2 (7.90%) <br> Cysteine-rich secretory protein (7.22%) | [103] |
| | *Naja atra* | 21 | 47 | Phospholipase A2 (45.6%) <br> 3-Finger Toxins (41.4) <br> NGF-beta family (2.4%) <br> SV metalloproteinases (1.5%) | [126] |
| **Workflow 2 SDS-PAGE/in-gel protein digestion/LC-MS/MS** | *Protobothrops kelomohy* | 11 | 42 | SV metalloproteinases (40.85%) <br> SV serine protease (29.93%) <br> Phospholipase A2 (15.49%) <br> L-amino acid oxidase (3.87%) | [127] |
| | *Deinagkistrodon acutus* | 16 | 103 | Phospholipase A2 (30%) <br> C-type lectins (21%) <br> Antithrombin (17.8%) <br> Thrombin (8.1%) | [128] |
| **Workflow 3 2DGE/in gel digestion/LC-MS/MS** | *Agkistrodon contortrix* | 10 | 26 | Phospholipase A2 (50.1%) <br> Metalloproteinases (25.26%) <br> Protein C activator (8.87%) <br> Serine protease (5.85%) | [109] |
| **Workflow 4 Shotgun analysis** | *Bothrops leucurus* | 19 | 137 | Phospholipase A2 (33.66%) <br> L-amino acid oxidases (9.18%) <br> SV serine proteinases (14.46%) <br> SV metalloproteinases (12.92%) | [114] |
| | *Deinagkistrodon acutus* | 10 | 84 | SV metalloproteinases (31.7%) <br> SV serine proteinases (17.6%) <br> C-type lectins (17.6%) <br> Phospholipase A2 (4.7%) <br> 5′-nucleotidase (5.9%) | [116] |
| **Combination of workflows 2 + 3** | *Ovophis monticola* | 9 | 247 | SV metalloproteases (36.8%) <br> SV serine proteases (31.1%), <br> Phospholipase A2 (12.1%) <br> L-amino acid oxidase (5.7%) | [129] |
| **Combination of workflow 1 + 2** | *Hydrophis schistosus* | 10 | 42 | Phospholipase A2 <br> Three-finger toxins | [112] |
| | *Naja naja* | 17 | 115 | Three-finger toxins (29%) <br> Phospholipase A2 (10%) <br> SV metalloproteinases (9%) | [44] |

## 4. Conclusions

Protein separation techniques are the basis on which proteomics analysis rely; thus, the choice of the convenient method is critical to reach the desired goal. Even with the scientific advancement, there is still no standard technique to be employed for the separation of SVs. Therefore, separation and analytical methods should be carefully chosen based on the objectives of research and the available resources. Clearly, snake venomics is of interest not only for fundamental research but also for the therapeutic field. It is true that venoms are toxic; however, they were shown to be an invaluable library for the development of pharmaceuticals. Thus, SV separation techniques play a pivotal role in the isolation and purification of biologically active molecules that could be used as model to develop drugs specifically to treat cardiovascular and neurological diseases. Another important field of venom studies is anti-venomics. Multiple whole proteome analysis techniques might

be used including 2-DGE, immunoaffinity chromatography and RP-HPLC to assess the immune reactivity of the antivenom to each component of the venom and to evaluate cross-reactivity with other species, altogether aiming to improve the specificity of antivenoms and reduce snakebite-related complications and mortalities [96].

**Author Contributions:** Conceptualization, Z.F.; validation, M.R., Z.F., C.M., J.-M.S. and C.L.; writing—original draft preparation, C.S.; writing—review and editing, M.R., C.M., Z.F. and C.L.; visualization, M.R. and C.S.; supervision, Z.F., C.M. and C.L.; project administration, Z.F. and C.L. All authors have read and agreed to the published version of the manuscript.

**Funding:** This work was supported by grants from the PHC CEDRE (46541TH) and the French MENESR. C.S is recipient of doctoral fellowship award from the Lebanese University in cooperation with the federation of Zgharta casa municipalities and also recipient of the Eiffel scholarship from Campus France.

**Institutional Review Board Statement:** Not applicable.

**Informed Consent Statement:** Not applicable.

**Data Availability Statement:** Not applicable.

**Acknowledgments:** We would like to thank Charbel Mouawad and Jacinthe Frangieh for the helpful discussion.

**Conflicts of Interest:** The authors declare no conflict of interest.

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
