# Peer review of "Separation and Analytical Techniques Used in Snake Venomics: A Review Article"

_processes, doi:10.3390/pr10071380_

Round 1

Reviewer 1 Report

The review is well written and offers a considerable numeber of informations. References are detailed.

Author Response

We thank Reviewer 1 for peer reviewing our manuscript and for her/his conclusion.

Reviewer 2 Report

Within the manuscript of “Separation and analytical techniques used in snake venomics: A review article”. Methodologies for the separation and analysis of snake venomics have been reviewed. The quality of presentation is good. Therefore, I would recommend this article for the publishing in the Process, but after the only one following point has been addressed:

 1.       Line 37, the reference shall be cited for the sentence of “The death toll is estimated to reach 138,000 deaths per year (WHO)”.

Author Response

Comments/question of reviewer 2  :

Within the manuscript of “Separation and analytical techniques used in snake venomics: A review article”. Methodologies for the separation and analysis of snake venomics have been reviewed. The quality of presentation is good. Therefore, I would recommend this article for the publishing in the Process, but after the only one following point has been addressed:

 1.       Line 37, the reference shall be cited for the sentence of “The death toll is estimated to reach 138,000 deaths per year (WHO)”.

Answer:

We thank Reviewer 2 for peer reviewing our manuscript and these construtive comments. As requested, this citation was added in the text.

Line 38: “ The death toll is estimated to reach 138,000 deaths per year (https://www.who.int/news-room/fact-sheets/detail/snakebite-envenoming accessed on January 2022).”

Reviewer 3 Report

The manuscript entitled “Separation and analytical techniques used in snake venomics: a review article” focused on the separation of snake venoms, which are composed of proteins and polypeptides. Authors should come up with new insights into the manuscript other than the following two articles. Most of the contents of this manuscript already covered these articles.

1.     Julien Slagboom, Chiel Kaal, Arif Arrahman, Freek J. Vonk, Govert W. Somsen, Juan J. Calvete, Wolfgang Wüster, Jeroen Kool, Analytical strategies in venomics, Microchemical Journal,Volume 175, 2022, https://doi.org/10.1016/j.microc.2022.107187.

2.     Abd El-Aziz, T. M., Soares, A. G., & Stockand, J. D. (2020). Advances in venomics: Modern separation techniques and mass spectrometry. Journal of chromatography. B, Analytical technologies in the biomedical and life sciences1160, 122352. https://doi.org/10.1016/j.jchromb.2020.122352

 The abstract does not reflect any findings. Authors should mention the major outcomes of the manuscript in the abstract as well. Regarding Figures 1, 2, 3, and 4, please indicate the matrix used in the particular column, washing buffer, and elution buffer widely employed in sake venoms research. You have used excessively unnecessary citations. For example, I did not see the usefulness of citations 13 to 23 under the subtopic ‘size-exclusion chromatography’. There is the same problem in other chromatographic techniques (section) as well. Thus, I suggest selecting the proper and most significant citations in the manuscript. I do not think Figure 5 is useful. You should present a schematic diagram for bioassay-guided fractionation. There are lots of grammatical issues in the manuscript. Some issues of langue are shown in the attached file.                                                                         

Author Response

Answer to reviewer 3 :

We thank Reviewer 3 for peer reviewing our manuscript and for helping us to improve our manuscript.

Point 1 : "The manuscript entitled “Separation and analytical techniques used in snake venomics: a review article” focused on the separation of snake venoms, which are composed of proteins and polypeptides. Authors should come up with new insights into the manuscript other than the following two articles. Most of the contents of this manuscript already covered these articles.
1.     Julien Slagboom, Chiel Kaal, Arif Arrahman, Freek J. Vonk, Govert W. Somsen, Juan J. Calvete, Wolfgang Wüster, Jeroen Kool, Analytical strategies in venomics, Microchemical Journal,Volume 175, 2022, https://doi.org/10.1016/j.microc.2022.107187.
2.     Abd El-Aziz, T. M., Soares, A. G., & Stockand, J. D. (2020). Advances in venomics: Modern separation techniques and mass spectrometry. Journal of chromatography. B, Analytical technologies in the biomedical and life sciences, 1160, 122352. https://doi.org/10.1016/j.jchromb.2020.122352

The abstract does not reflect any findings. Authors should mention the major outcomes of the manuscript in the abstract as well."

Answer: Thank you for these comments ! The main outcome of the review was that no technique could be considered as a ‘gold standard’ for the analysis of snake venoms but different combinations could be developed to reach the study objectives. This idea appeared in the abstract as suggested in the two following sentences:

Line 28-31 “Eventually, we show that to date, there is no standard technique used for the separation of all snake venoms. Thus, different combinations might be developed taking into consideration the main objective of the study, the available resources, and the properties of the target molecules to be isolated.”

Point 2 : "Regarding Figures 1, 2, 3, and 4, please indicate the matrix used in the particular column, washing buffer, and elution buffer widely employed in sake venoms research."

Answer: The most commonly used matrices and buffers for SV separation were mentionned in the captions of figures 1 to 4 (pages 3, 5, 6, and 7).

Point 3: "You have used excessively unnecessary citations. For example, I did not see the usefulness of citations 13 to 23 under the subtopic ‘size-exclusion chromatography’. There is the same problem in other chromatographic techniques (section) as well. Thus, I suggest selecting the proper and most significant citations in the manuscript."

Answer: References have been revised and reduced from 144 to 122 to keep the most relevant and recent ones.

Point 4 : "I do not think Figure 5 is useful. You should present a schematic diagram for bioassay-guided fractionation."

Answer: Figure 5 was removed from the manuscript and substituted by a diagram for bioassay-guided fractionation. Please refer to the manuscript page 8.  

Point 5: "There are lots of grammatical issues in the manuscript. Some issues of langue are shown in the attached file."
Answer: Thank you for your corrections. We used your file to fix all grammatical issues and we followed all your comments.

Reviewer 4 Report

The review is systematically designed, and the results are meaningful. However, some parts should be made clear, and the ambiguities should be improved. Please see the comments below.

1) 2.1. Gel-free methods. I suggest changing the title; everything included here refers to chromatographic techniques.

2) Error in Figure 2A` Says Negatively charged protein flows through

and must say “Positively charged protein flows through”

 3) I would have liked to see in each paragraph referring to chromatographic methods of these chromatographic techniques applied in the separation or fractionation of SV. Unfortunately, the authors only give a few examples. I believe a more in-depth discussion of the results should be given.

4) Basic literature that should be removed is provided at the beginning of each paragraph. for instance. Lines from 84 to 94 (Size exclusion chromatography…..by size only [15]), lines from 130 to 154 (Ion exchange chromatography…..different pH values [24]),  lines from 180 to 188 (Affinity chromatography……time-consuming method [33]), lines 225 to 230, lines from 268 to 273, lines from 295 to 302.

5) Line 325. Vipera ammodyes ammodytes venom is correct?

6) Remove line 353-354 since they talk about information that has not been published.

“Recently, we took advantage of zebrafish embryos transparency to assess direct cardiotoxicity of SV (paper submitted to Biology, MDPI)”

7) Table 1. Serine protease  Bothrops jararaca  28,000 Da; while that in  Crotalus durissus collilineatus  29474 Da. Please homogenize.

8) 3.2. Whole proteome characterization and identification.

This section will need to be reduced. In this section, the authors provide much information regarding the technique and give very little information regarding the studies where it has been applied. I suggest removing non-essential information and focusing the discussion and analysis on the results previously obtained. In lines 385 to 409, this information is not necessary. Besides, the information provided in lines 468 to 477 is relevant to the purpose of this review.

 9) Indicate what means panel A, C, C, and D are in Figure 6. Add this information to the figure legend.

 10)Probably a table of advantages and disadvantages of each technique could be made

 11) Conclusion. Lines 504 to 506 is not related with the aim to this review.

Author Response

Dear Reviewer 4,

We thank you for peer reviewing our manuscript. Below, we answered to your questions and comments in eleven points.

“The review is systematically designed, and the results are meaningful. However, some parts should be made clear, and the ambiguities should be improved. Please see the comments below.”

Point 1. “ 2.1. Gel-free methods. I suggest changing the title; everything included here refers to chromatographic techniques.”

Answer: We agree with this point. Indeed, this section refers to chromatographic techniques and ‘gel-free techniques’ was not well choosen. We substituted it with the term ‘chromatographic techniques’ Similarly, the term’ gel-based techniques’ was substituted by ‘electrophoretic techniques’. Thanks for this comment.

Point 2. “Error in Figure 2A` Says Negatively charged protein flows through and must say “Positively charged protein flows through””

Answer: “Negatively” was changed to “positively” in figure 2A’. Please refer to manuscript page 5.

 Point 3. “I would have liked to see in each paragraph referring to chromatographic methods of these chromatographic techniques applied in the separation or fractionation of SV. Unfortunately, the authors only give a few examples. I believe a more in-depth discussion of the results should be given.”

Answer: Thanks for your comment. Thus, in each chromatographic techniques, the examples that were given were further developed and discussed. For size exclusion chromatography, we explained the use of this technique as a first step of separation, indicating its usefulness to separate molecules by size groups and highlighting on its low resolution throughout the references given (Please see lines “199-205”).

For ion exchange chromatography, we elaborated more on the examples given for CEX and AEX to better explain their usage in SV separations(Please see lines “254-272”).

For RP-HPLC, we discussed our examples to show the effectiveness and versatility of HPLC use (Please see lines “455-492”, section 2.1.4.).

Point 4. “Basic literature that should be removed is provided at the beginning of each paragraph. for instance. Lines from 84 to 94 (Size exclusion chromatography…..by size only [15]), lines from 130 to 154 (Ion exchange chromatography…..different pH values [24]),  lines from 180 to 188 (Affinity chromatography……time-consuming method [33]), lines 225 to 230, lines from 268 to 273, lines from 295 to 302.”

Answer: As requested, the basic information in the beginning of each sections (sections 2.1.1 to 2.2.2) were removed. These changes appeared thanks to the tracking mode of word in our revised manuscript.

Point 5. “Line 325. Vipera ammodyes ammodytes venom is correct?”

Answer: Yes, it was a mistake. The name of this snake was corrected.

Point 6. “Remove line 353-354 since they talk about information that has not been published. “Recently, we took advantage of zebrafish embryos transparency to assess direct cardiotoxicity of SV (paper submitted to Biology, MDPI).”

Answer: Our article has been accepted for publication in Biology (doi: 10.3390/biology11060888) during the revision of this review. The reference of this article is now incremented in the text and in the reference list (lines 698-699 reference [55]).

Point 7. “Table 1. Serine protease Bothrops jararaca 28,000 Da; while that in Crotalus durissus collilineatus 29474 Da. Please homogenize.”
Answer: All molecular weights (MW) in the table have been written with a comma to separate groups of thousands. We added in this table an indication concerning the technique used to determined MW of each compound: MW determined by SDS-PAGE (a) and by mass spectrometry (b). This table 1 became Table 2 in the revised manuscript (see point 10).

Point 8. “3.2. Whole proteome characterization and identification. This section will need to be reduced. In this section, the authors provide much information regarding the technique and give very little information regarding the studies where it has been applied. I suggest removing non-essential information and focusing the discussion and analysis on the results previously obtained. In lines 385 to 409, this information is not necessary. Besides, the information provided in lines 468 to 477 is relevant to the purpose of this review.”
Answer: As requested, we reduced this section and removed non-essential information to keep only relevant information. Thanks for this comment.

 Point 9. “Indicate what means panel A, C, C, and D are in Figure 6. Add this information to the figure legend.”
Answer: The caption of Figure 6 (page 13) was completed as requested.

 Point 10. “Probably a table of advantages and disadvantages of each technique could be made.”
Answer: Thanks for this comment. We propose a new table (Table 1 in the revised manuscript) which describes the advantages and disadvantages of each technique (please see  page 9).

 Point 11. “Conclusion. Lines 504 to 506 is not related with the aim to this review.”
Answer: That is right. We removed the two first sentences of our conclusion (“The evolution of Omics technologies... the venom’s evolution). As requested by reviewer 5, we completed our conclusion with four new sentences bringing perspectives of Omic techniques in regard to drug development and health related issues.

Sincerely yours

Christian Legros

Reviewer 5 Report

In this manuscript, the authors reported that the summary of numerous separation and analytical techniques for snake venom composition including their applications in snake venom research. In addition, the different possible technical combinations that could be used to separate venoms were reported too. The authors also described four different workflows and different proteomic strategies that could be applied for future venomic studies. In overall, this manuscript is interesting but in order to consider publication, this work should be revised. The following comments should be addressed for the improvement of their manuscript.

Comment 1: The overall study aims for this review study about the advanced separation and analytical techniques used in snake venomics need to be further clarified in detail as compared to traditional/conventional techniques.

Comment 2: The various recent reports and their research findings on the “various methods used for separation of venom complex mixtures” should be summarized into a table form and discussed for better understanding in term of effectiveness, duration, etc.

Comment 3: The future direction and perspectives for these advanced separation and analytical techniques used in snake venomics study can be further discussed in the conclusion section. The detail of therapeutic field in-512 cluding anti-venom and SV-derived drugs development can be further elaborated.

Comment 4: The carefully English correction is necessary for the whole manuscript. Please check and revise accordingly.

Author Response

Dear Reviewer 5,

We thank Reviewer 5 for peer reviewing our manuscript and for helping us to improve our manuscript. We answered to your questions and comments below.

“In this manuscript, the authors reported that the summary of numerous separation and analytical techniques for snake venom composition including their applications in snake venom research. In addition, the different possible technical combinations that could be used to separate venoms were reported too. The authors also described four different workflows and different proteomic strategies that could be applied for future venomic studies. In overall, this manuscript is interesting but in order to consider publication, this work should be revised. The following comments should be addressed for the improvement of their manuscript.”

Comment 1: “The overall study aims for this review study about the advanced separation and analytical techniques used in snake venomics need to be further clarified in detail as compared to traditional/conventional techniques.”

Answer: Thank you for your comments. In fact, the main objective of this review was to summarize current knowledge about separation techniques used in snake venom analysis. Therefore, we first summarized the currently separation techniques and then we discussed their implementation in snake venom analysis. To this end, we reviewed the use of separation techniques to fulfil distinct study objectives:

  1. Objective 1: The use of chromatographic techniques like SEC, IEX and RP-HPLC to isolate a protein using bioassay-guided fractionation (what is known as conventional techniques)
  2. Objective 2: The use of different separation techniques (like RP-HPLC, 1-DGE, 2-DGE) to evaluate whole venom proteomic composition (snake venomics) using the advanced analytical techniques

Therefore, the abstract was modified, as well the last part of the introduction to make these two objectives more explicit and precise. At the end of the introduction, we added the following sentences:

 “Accordingly, we present in this review the most used separation techniques currently available for the analysis of snake venoms. We then, detail their implementation and use for different study purposes. In fact, it is the study objective that drives the choice of separation technique to be used. Therefore, we detail at first in this review the combinations of chromatographic techniques that are used to isolate and purify different snake venom proteins. On the other hand, we detail different workflows using different separation techniques for the evaluation of whole snake venom proteomic composition and distribution of SV protein families.”

Comment 2: “The various recent reports and their research findings on the “various methods used for separation of venom complex mixtures” should be summarized into a table form and discussed for better understanding in term of effectiveness, duration, etc.”
Answer: Numerous recent reports have been summarized in table 2 (ex-table 1) to numerate different types of chromatographic techniques and various combinations that can be used to isolate and purify snake venom proteins belonging to different protein families.

As for the separation techniques used for proteomics analysis, multiple reports were summarized in table 3 (ex table 2) showing the protein families detected for each snake venom, the number of protein families and number of proteins.

Unfortunately, it is complicated to compare all these techniques in terms of effectiveness and duration, since important information has not been given in the articles. Of course, we think that such comparison and analysis would be helpful for the readership.

Comment 3: “The future direction and perspectives for these advanced separation and analytical techniques used in snake venomics study can be further discussed in the conclusion section. The detail of therapeutic field in-512 cluding anti-venom and SV-derived drugs development can be further elaborated.”

Answer: We changed the conclusion as requested. We completed our conclusion with four new sentences bringing perspectives of Omic techniques in regard to drug development and health related issues. We added the following sentences:

“It is true that venoms are toxic, however, they were shown to be an invaluable library for the development of pharmaceuticals. Thus, SV separation techniques play a pivotal role in the isolation and purification of biologically active molecules that could be used as model to develop drugs specifically to treat cardiovascular and neurological diseases. Another important field of venom studies is anti-venomics. Multiple whole proteome analysis techniques might be used including 2-DGE, immunoaffinity chromatography and RP-HPLC to assess the immune-reactivity of the antivenom to each component of the venom and to evaluate cross-reactivity with other species altogether aiming to improve the specificity of antivenoms and reduce snakebite-related complications and mortalities [91].“

Comment 4: “The carefully English correction is necessary for the whole manuscript. Please check and revise accordingly.”

Answer: We read carefully the manuscript and we corrected English spelling and grammar errors. This could be appreciated in the revised manuscript.

Sincerely yours

Christian Legros

Round 2

Reviewer 3 Report

Dear authors,

The manuscript should cite the related previous work such as      https://doi.org/10.1016/j.microc.2022.107187; and https://doi.org/10.1016/j.jchromb.2020.122352.  

Although you have mentioned matrices and buffers for SV separation in the captions of figures 1 to 4, please try to improve them by incorporating factful examples from literature. Alternatively, you can cite some published images (figures) in SV separation after obtaining reproduction license. In my option, figure 1-4 are too ordinary as to classroom teaching. 

Author Response

Dear Reviewer,

Thanks again for taking time with our manuscript and your comments. We revised the manuscript according your indications. The responses to each point are detailed below.

Point 1. The manuscript should cite the related previous work such as      https://doi.org/10.1016/j.microc.2022.107187; and https://doi.org/10.1016/j.jchromb.2020.122352.  

The reference https://doi.org/10.1016/j.microc.2022.107187 was added in section 3.2 line 444. We moved the reference 96 (ex-91) (doi:10.1186/s40409-017-0117-8) from line 438 to the same place. Concerning the second reference (https://doi.org/10.1016/j.jchromb.2020.122352), it has been already cited in the introduction but at the wrong place:

Lines 49-50 “Also, it will pave the way for the discovery of novel biomolecules with therapeutic interest [4].” We changed reference 4 by this one : DOI: 10.3390/toxins11100564, which corresponds to the topic of this sentence. Sorry for this mistake. We transfered the reference (https://doi.org/10.1016/j.jchromb.2020.122352) to section 3.2 at the end of the sentence line 438.

Point 2. Although you have mentioned matrices and buffers for SV separation in the captions of figures 1 to 4, please try to improve them by incorporating factful examples from literature. Alternatively, you can cite some published images (figures) in SV separation after obtaining reproduction license. In my option, figure 1-4 are too ordinary as to classroom teaching. 

As to your suggestion, we modified figures 1,2 and 4 to include examples from literature of some proteins that have been isolated using these chromatographic strategies. Thus, SEC, IEX and RP-HPLC figures were all merged into one figure since they can be used in combination to isolate different snake proteins.

As for figure 3, since affinity chromatography is a bit different from the other techniques, we kept the figure apart. However, we added another panel to the figure (panel C) showing a chromatogram of eluted proteins to better explain the equilibration and elution steps during this process.

We hope that this revised version would be suitable for publication,

Kind regards,

Christian Legros

Ziad Fjloun

Reviewer 5 Report

In overall, this manuscript was technically well revised. This revised manuscript meets the criteria of Processes. Therefore, in my opinion, the revised manuscript can be accepted for publication.

Author Response

dear Reviewer,

thanks a lot for your peer-reviewing.

kind regards

Christian Legros

Ziad Fajloun